# Spatiotemporal Mapping of Grazing Livestock Behaviours Using Machine Learning Algorithms

**DOI:** 10.3390/s25154561

**Published:** 2025-07-23

**Authors:** Guo Ye, Rui Yu

**Affiliations:** 1School of Ecology, Hainan University, Haikou 570228, China; yeggis@outlook.com; 2Department of Geography, The University of Manchester, Manchester M13 9PL, UK

**Keywords:** livestock behaviours, vehicle GPS sensor, machine learning classification, spatial clustering, temporal probability distribution

## Abstract

Grassland ecosystems are fundamentally shaped by the complex behaviours of livestock. While most previous studies have monitored grassland health using vegetation indices, such as NDVI and LAI, fewer have investigated livestock behaviours as direct drivers of grassland degradation. In particular, the spatial clustering and temporal concentration patterns of livestock behaviours are critical yet underexplored factors that significantly influence grassland ecosystems. This study investigated the spatiotemporal patterns of livestock behaviours under different grazing management systems and grazing-intensity gradients (GIGs) in Wenchang, China, using high-resolution GPS tracking data and machine learning classification. the K-Nearest Neighbours (KNN) model combined with SMOTE-ENN resampling achieved the highest accuracy, with F1-scores of 0.960 and 0.956 for continuous and rotational grazing datasets. The results showed that the continuous grazing system failed to mitigate grazing pressure when grazing intensity was reduced, as the spatial clustering of livestock behaviours did not decrease accordingly, and the frequency of temporal peaks in grazing behaviour even showed an increasing trend. Conversely, the rotational grazing system responded more effectively, as reduced GIGs led to more evenly distributed temporal activity patterns and lower spatial clustering. These findings highlight the importance of incorporating livestock behavioural patterns into grassland monitoring and offer data-driven insights for sustainable grazing management.

## 1. Introduction

Grassland ecosystems are vital for livestock production, biodiversity conservation, and ecosystem regulation, with grazing being a dominant land-use activity [1,2] Different livestock behaviours directly influence grassland ecosystems through various mechanisms [3]. For example, grazing behaviour influences the pasture structure and composition, which determines the plant growth and recovery capacity; lying and standing behaviour affects vegetation and soil physiology and biochemistry, influencing soil stability and nutrient cycling (e.g., concentrated excreta distribution); and walking behaviour impacts the soil structure, vegetation utilisation rate, and pasture growth [4,5,6,7].

Variations in grazing-intensity gradients (GIGs) and grazing management systems are key drivers of selective grazing behaviours, which ultimately shape the spatial structure and condition of grassland growth. Changes in GIGs can alter livestock movement patterns and determine the extent of pasture utilisation by affecting the spatiotemporal availability of foraging resources [8]. At the behavioural level, GIGs have been shown to significantly influence the intensity and frequency of core livestock activities, such as grazing, resting, and walking [3]. Moreover, different grazing management regimes—namely rotational grazing and continuous grazing—regulate the temporal distribution of these behaviours, thereby modifying defoliation patterns and the degree of disturbance imposed on vegetation [9]. Therefore, uncontrolled and excessive grazing can decrease ecosystem stability and productivity, reducing available forage resources, and ultimately lead to severe grassland degradation [10,11].

Fine-scale livestock behaviour tracking and spatiotemporal analysis are essential for sustainable grassland management. In the early stages, grazing management relied heavily on direct visual observation of livestock behaviour and vegetation defoliation levels; however, this approach is both time-consuming and labour-intensive. Moreover, livestock generally exhibit limited movement distances, which are accompanied by complex and dynamic behavioural patterns. In areas with heterogeneous landscapes and complex terrain, conducting long-term, systematic observations using the naked eye becomes increasingly challenging. In recent years, Global Positioning System (GPS) tracking has become a powerful tool for monitoring livestock movements. Numerous studies have employed GPS techniques to reveal spatial behaviour patterns. For example, Cristescu [12] examined mining impacts on grizzly bear movement in Alberta, Canada, using 9 years of GPS data comparing movement patterns during active mining and after mine closure, using indicators, such as home range overlap, step length, selection ratio, and the ratio of steps within the disturbance boundary to steps crossing the disturbance, to analyse grizzly bears’ response patterns to mining activities. Millward et al. [8] used GPS collars to track cattle location across seven ranches under two adjustment factors: distance to water and slope. Gou et al. [13] used GPS tracking and field biomass measurements to analyse cattle foraging patterns on different terrains (lowland vs. dunes) from July to September. They found that the average foraging density was significantly higher in lowlands, increasing from 0.61 in July to 0.66 in August and 0.88 in September. Foraging density in lowlands was negatively correlated with biomass (*p* = 0.07), whereas no significant correlation was observed on dunes, suggesting that higher cattle aggregation in lowlands leads to greater vegetation consumption. To further understand how livestock activities respond to the environment and impact vegetation dynamics, GPS data have increasingly been integrated with Geographic Information Systems (GIS) and remote sensing techniques. Kawamura et al. [14] used a grid-based analysis to examine the correlation between grazing intensity, as derived from GPS-tracked movements of three sheep herds, MODIS NDVI, and field-measured biomass. The study also demonstrated that areas with higher grazing intensity exhibited more severe vegetation degradation, confirming the negative impact of concentrated grazing on plant productivity. Wang et al. [15] utilised GPS collars to track the spatiotemporal distribution of sheep under different grazing-intensity gradients in summer and autumn. By integrating the GPS data with remote sensing information on terrain and vegetation, the study revealed significant seasonal and terrain-related variations in movement distances and foraging preferences. Moreover, previous studies have shown that in free-range grazing systems, especially in heterogeneous mountainous or sandy landscapes, GPS tracking is effective in overcoming the limitations of coarse management practices and the challenges of direct behavioural monitoring [16,17,18]. In contrast, within enclosed grazing systems, GPS enables more precise and adaptive management—for instance, by supporting dynamic stocking rate adjustments and facilitating quantitative analysis of the causal relationships between grazing regimes, intensity gradients, and livestock behavioural patterns. However, due to the growing volume and complexity of GPS-based behavioural datasets, traditional analytical approaches are often insufficient. In response, machine learning (ML) models have been introduced as efficient tools for extracting behavioural patterns and revealing the spatiotemporal mechanisms of selective grazing.

Machine learning (ML) approaches, including feature selection [19], clustering [20], and regression/classification [21], have gained significant attention, particularly in the field of grazing research where classification algorithms are most commonly used. For example, Augustine et al. [22] collected GPS and activity sensor data from semi-arid rangelands in eastern Colorado between 2008 and 2010 and applied a Classification and Regression Tree (CART) model to distinguish between grazing and non-grazing activities, achieving a relatively low misclassification rate of 12.9%. However, the binary nature of decision trees led to reduced performance when classifying multiple behavioural categories. To overcome such limitations, ensemble learning methods, such as Random Forest and XGBoost, have been increasingly adopted. Williams et al. [23] evaluated the performance of four machine learning algorithms—naïve Bayes, JRip, J48, and Random Forest—for dairy cow behaviour recognition and identified JRip as the most effective under a 32-segment strategy, with an overall classification accuracy of 0.85. Homburger et al. [3] further explored spatial activity patterns of dairy cows across six summer grazing areas using Random Forest for behaviour classification and regression modelling to assess environmental and management impacts. The study highlighted terrain slope, forage quality, stocking rate, and rotational grazing duration as major determinants of cattle behavioural distribution.

In summary, based on the reviewed literature, several key research gaps can be identified:

Class imbalance in GPS data: Previous studies have identified that GPS data for livestock behaviour often suffer from class imbalance, particularly with certain behaviours, like grazing, dominating the dataset. This imbalance creates prediction bias in machine learning classifiers. However, to date, limited research has systematically examined the effects of different resampling strategies on the performance of various classifiers across different grazing management datasets.

Spatial clustering of behaviours: One key research gap is which behavioural patterns are susceptible to being driven to form hotspots and how the spatial clustering of these behaviours varies in response to different grazing-intensity gradients and grazing management strategies.

Probability distribution of grazing time: Limited attention has been given to the temporal dynamics of livestock behaviours. Specifically, how the proportion of time spent on key livestock behaviours and the temporal concentration of grazing behaviour vary with changes in grazing-intensity gradients and management systems remains unclear.

To address these gaps, this study conducted grazing experiments using two grazing management strategies: rotational and continuous grazing each with three grazing-intensity gradients, namely heavy, moderate, and light, using GPS collars to track cattle movement. To identify the optimal classifier for accurately classifying five key cattle behaviours, four representative machine learning algorithms were tested in combination with five data resampling strategies. This optimal model was then used to efficiently analyse the spatiotemporal patterns of these behaviours in response to varying GIGs and grazing management systems.

## 2. Material and Methods

### 2.1. Study Area

The study was conducted in Nanba Village, Wenchang City, located in the northeastern part of Hainan Island, China (approximately 110° E, 19° N) (Figure 1). This region is characterised by a mixed agroecosystem in which both crop cultivation and enclosed livestock farming are practiced. The elevation values, derived from a Digital Surface Model (DSM), range from −1.79 m to −0.85 m, indicating slight topographic variation and a relatively low-lying terrain. The area has a tropical monsoon island climate, marked by distinct wet and dry seasons. The rainy season typically spans from May to October, while the dry season lasts from November to April. Annual average temperatures range between 22.5 °C and 25.6 °C, and mean annual precipitation varies from 900 mm to 2500 mm. The region’s warm, humid conditions and relatively high rainfall provide abundant water and favourable climatic conditions for livestock production. In recent years, due to poor management of artificial grassland grazing, an overemphasis on livestock quantity, and a lack of sustainable grassland investment, exploitative grazing practices have led to significant grassland degradation.

To evaluate the effects of grazing management systems and grazing intensities on the spatiotemporal distribution of livestock behaviours, three paddocks (A, B, and C) were established within the study area. The paddocks were subjected to a decreasing gradient of grazing intensity under both rotational and continuous grazing systems. The areas of paddocks A, B, and C were approximately 1100 m^2^, 1100 m^2^, and 1700 m^2^, respectively. The corresponding average stocking rates were set at 41 head/hm^2^ for heavy-intensity grazing, 23 head/hm^2^ for moderate-intensity grazing, and 13 head/hm^2^ for light-intensity grazing. This design was intended to simulate grazing intensity gradients commonly investigated in larger-scale temperate grassland studies, while being adapted to the constraints of tropical fenced pastures and short-term observation feasibility in Hainan. The grazing intensity levels were established based on the analysis of tropical pasture grazing intensities by Costa et al. [24] and informed by the local production practice, which typically involves a stocking rate of approximately 25 head per hectare. In particular, we considered stocking rate (*SR*), paddock area (*PA*), and grazing duration (*T*) as three interrelated factors that jointly define grazing pressure (*GP*), as expressed in Equation (1):(1)$$GP=\frac{SR*T}{PA}$$

By integrating these parameters in a balanced design, we aimed to ensure experimental comparability and to effectively evaluate the combined effects of grazing management system and grazing intensity on the spatial clustering and temporal distribution of cattle behaviours.

### 2.2. Grazing Management and GPS

According to Kilgour et al. [25], cattle can exhibit up to 40 distinct behaviours, although many of these are rare and occur only for short durations. Based on the reviewed literature on livestock behaviour, most studies typically focus on either the distinction between foraging and non-foraging activities or select a limited number—often three—of key behaviours for analysis. Therefore, based on the ecological significance of different behaviours, their relative impact on vegetation, and their frequency of occurrence in grazing cattle, this study identified and analysed five primary behaviours: grazing, rumination, standing, lying, and walking (Figure 2). Grazing experiments were conducted daily from 8:30 a.m. to 5:30 p.m.—a timeframe selected to align with the conventional schedule of local pastoral livestock management systems. Artificial behavioural observations were made during the pre-grazing phase of each experiment. Meanwhile, GPS tracking was implemented during three grazing intensity gradients: Heavy grazing was conducted from 10–19 November 2022, involving 9 animal units (AU; 1 AU = 250 kg adult cattle) in the rotational grazing areas (A and B) and 7 AU in continuous grazing area C, with an average stocking rate of 41 AU/ha. Moderate grazing occurred from 9–18 December 2022, with 5 AU in A and B and 4 AU in C, resulting in an average stocking rate of 23 AU/ha. Light grazing was carried out from 20–29 December 2022, with 3 AU in A and B and 2 AU in C, corresponding to an average stocking rate of 13 AU/ha. In the rotational grazing system, A was grazed for 5 days, followed by 5 days in B, whereas the continuous grazing area C was grazed uninterrupted for 10 days during each grazing intensity phase.

The GPS devices used in this study were Q8 high-frequency vehicle-mounted units ((P3-A, Shenzhen, Guangdong, China)) with an adjustable sampling interval of 10 s, each integrated with an MT2503D chip supporting both GPS and BeiDou satellite navigation systems. Real-time position data were transmitted via the GPRS network, facilitated by multiple antenna components, including a GPS antenna, GSM antenna, ceramic antenna, and FPC antenna, which ensured stable signal reception under field conditions. Additionally, each device incorporated a three-axis accelerometer (G-sensor) to detect motion status, enabling energy-efficient data collection by switching to standby mode during periods of inactivity and resuming recording once movement was detected. The device dimensions were 58 × 32.5 × 20.8 mm, with a mainboard size of 47.5 × 24 × 4.8 mm and a total weight of 146.94 g. Each device featured three indicator lights: a blue light for GPS signal status, a green light for network signal status, and a red light for charging status. The battery capacity was 5000 mAh, and the positioning accuracy was within 1 m. To ensure precise tracking of cattle movements, each GPS device was first placed at a location with known coordinates. Positional errors were identified by comparing the recorded coordinates from the collar with the reference location, and calibration was conducted accordingly. Each GPS unit was then secured inside a waterproof Oxford cloth collar (11.5 × 6.5 × 4 cm). To enhance the resolution of tracking data, each cow was equipped with two GPS devices throughout the day, enabling differential GPS correction (DGPS) and achieving an average update interval of approximately 5 s. Each grazing period involved equipping approximately half of the total cattle population with GPS collars. The GPS devices remained in standby mode when the cattle were stationary to conserve power and resumed recording upon movement detection. Only healthy individuals of comparable body size were selected to minimise behavioural variability, and each animal was continuously monitored using two GPS collars under both rotational and continuous grazing conditions. Importantly, the total weight of the GPS collar system (146.94 g) was well within the threshold deemed non-intrusive for livestock behaviour monitoring. Prior studies have established that biologging devices exert negligible influence on animal activity patterns when their weight remains below a critical proportion of the animal’s body mass [26,27,28]. Throughout the observation period, no behavioural anomalies attributable to the device were observed, further confirming its suitability for use in naturalistic grazing environments.

### 2.3. Data Sets

Before the formal commencement of the experiment, all observers underwent structured training in collaboration with experienced local herders. The herders shared their practical expertise in identifying characteristic body postures and movement cues associated with key cattle behaviours, including grazing, walking, standing, resting, and ruminating. Grazing was characterised as the activity of searching for and consuming forage. Rumination referred to the regurgitation and repeated chewing of previously swallowed feed. Lying behaviour was described as resting on the ground without performing any additional actions. Walking was defined as a continuous forward motion with the animal’s head held upright and no associated jaw activity, while standing referred to a static posture with an upright head and no jaw movement. This preparatory training enabled observers to accurately and independently classify and record cattle behaviours in a consistent manner throughout the observation period. Simultaneously, the GPS collar devices automatically recorded spatiotemporal data at 5-s intervals, including date, time, longitude, latitude, and velocity. All GPS coordinates (latitude and longitude), encoded timestamps, and instantaneous speed values were used as predictor variables and input into the machine learning model.

### 2.4. Data Preprocessing and Classifier Learning

A significant class imbalance was observed in the dataset, with approximately 80% of the data corresponding to grazing behaviour, while the remaining four behaviours accounted for only 20% of the total data. This imbalance led to a high overall classification accuracy but poor generalisation performance, with a strong bias toward predicting the majority class while underrepresenting behaviours such as rumination and lying. To mitigate the issue of class imbalance commonly observed in livestock behaviour datasets, particularly those dominated by grazing records, five resampling strategies were systematically evaluated. These included: (1) combined sampling using the Synthetic Minority Oversampling Technique and Edited Nearest Neighbours (SMOTE-ENN), (2) combined sampling using SMOTE and Tomek Links (SMOTE-Tomek), (3) oversampling using the Synthetic Minority Oversampling Technique (SMOTE), (4) oversampling using Adaptive Synthetic Sampling (ADASYN), and (5) undersampling using Cluster Centroids. Each strategy was applied to enhance the representativeness of minority behaviour classes and reduce classifier bias. This evaluation provides insights into the relative effectiveness of resampling methods under different grazing management systems and supports the development of more robust behaviour classification models.

Before training the classifier, a data cleaning process was conducted to remove noise and inconsistencies. The dataset was then randomly split into a training set and a testing set using a 7:3 ratio. Model performance was evaluated on the testing set using multiple metrics to ensure robust assessment of prediction accuracy. The four representative classification algorithms tested included Naïve Bayes (Probabilistic Generative Model), k-Nearest Neighbours (KNN) (Instance-Based Learning), Extreme Gradient Boosting (XGBoost) (boosting-based ensemble learning), and Random Forest (bagging-based ensemble learning). GPS-tracked location data and manually observed livestock behaviour data were incorporated as the training dataset input for the model. A stratified 10-fold cross-validation (10-FCV) method was used to evaluate classifier performance. In 10-FCV, the dataset was randomly divided into 10 equal subsets while maintaining the class distribution. Nine subsets were used for training, and the remaining subset was used for validation. This process was repeated until all 10 subsets had been used for both training and validation, allowing for the calculation of the average classification accuracy (CA) and error rate. Additionally, the process was repeated across 10 different randomised runs (10 × 10 FCV) to enhance the robustness of classifier learning. The best-performing classifier was subsequently applied to conduct behaviour prediction analyses using the real GPS trajectory data collected from two grazing management systems under decreasing grazing-intensity gradients.

To improve model adaptability to the specific characteristics of the cattle behaviour’s dataset, several algorithm-specific hyperparameters were tuned during the training phase. For the Naïve Bayes classifier, a Gaussian distribution was used as the prior, with the Laplace smoothing parameter (alpha) set to 1, and a binarization threshold of 0. In the Random Forest algorithm, the Gini index was employed as the criterion for node splitting. The maximum tree depth was set to 90 for the continuous grazing dataset and 145 for the rotational grazing dataset. The number of decision trees was set to 400 and 350 for the continuous and rotational systems, respectively. The minimum number of samples required to split an internal node was set to 6 under continuous grazing and 2 under rotational grazing, while the minimum number of samples required at a leaf node was set to 3 and 1, respectively. For the K-Nearest Neighbours (KNN) classifier, the Manhattan distance metric was selected, as it yielded better classification accuracy compared to the Euclidean and Chebyshev distances. The number of neighbours (K) was set to 2 for both grazing systems. In the XGBoost algorithm, the base learner was set to ‘gbtree’. The maximum tree depth was set to 9 under continuous grazing and 10 under rotational grazing. The learning rate was set to 0.07 and 0.1, respectively. The subsample ratio (i.e., the proportion of training samples used per tree) was 0.7 for continuous grazing and 0.8 for rotational grazing, while the colsample_bytree parameter (i.e., the fraction of features used per tree) was fixed at 1 in both cases.

### 2.5. Model Evaluation

A confusion matrix (Table 1) is a widely used tool for assessing classification model performance, particularly in supervised learning (Equation (1)). It provides a tabular representation of the relationship between true and predicted class labels, allowing for a detailed evaluation of model effectiveness. The key elements of the confusion matrix were:True Positive (TP): Correctly predicted positive instances.True Negative (TN): Correctly predicted negative instances.False Positive (FP): Incorrectly predicted positive instances.False Negative (FN): Incorrectly predicted negative instances.

Classification accuracy (*CA*) is commonly used as an evaluation metric, defined as the proportion of correctly classified instances:(2)$$CA=\frac{TP+TN}{TP+FP+FN+TN}$$

However, accuracy alone is not always a reliable indicator of model performance, especially in imbalanced datasets where the classifier may achieve high accuracy by predominantly predicting the majority class. To address this limitation, precision and recall were considered (Equations (2) and (3)):(3)$$Precision=\frac{TP}{TP+FP}$$(4)$$Recall=\frac{TP}{TP+FN}$$

Precision and recall often exhibit a trade-off, as increasing one typically decreases the other. To balance these metrics, the $F_{1}$-*score* was employed as a harmonic mean of precision and recall (Equation (4)):(5)$$F_{1}-\mathrm{score}=\frac{2*Precision*Recall}{Precison+Recall}$$

Additionally, the Kappa coefficient (κ) was calculated to measure classification consistency beyond chance, accounting for random agreement between predictions and true labels (Equations (5) and (6)). Kappa values range from −1 to 1, with higher values indicating better agreement, which represents the expected accuracy given the class distribution.(6)$$Expected\;Accuracy=\frac{(\sum \left(Row\;Total*\;Column\;Total)\right)}{{\left(Total\;Elements\right)}^{2}}$$(7)$$Kappa=\frac{CA-Expected\;Accuracy}{1-Expected\;Accuracy}$$

These evaluation metrics are used to determine the classification performance of the model and to select the optimal model.

### 2.6. Zonal Grid Statistics and Spatial Hotspot Analysis of Livestock Behaviours

To quantitatively evaluate the spatial distribution patterns of cattle behaviours under different grazing intensities, the entire study area—comprising three grazing management units (A, B, and C) with areas of 1100 m^2^, 1100 m^2^, and 1700 m^2^, respectively—was partitioned into grid cells using a fishnet approach. We tested three spatial resolutions (1 m × 1 m, 5 m × 5 m, and 10 m × 10 m) and found that only the 1 m × 1 m resolution was able to capture distinct and statistically meaningful spatial patterns, including non-significant zones and cold/hotspots at the 99%, 95%, and 90% confidence levels. Therefore, the 1 m × 1 m grid was adopted for subsequent spatial hotspot analysis to ensure sufficient spatial detail and statistical sensitivity. Within each grid, the number of GPS trajectory points corresponding to five cattle behavioural categories (as classified by a machine learning model) was aggregated to generate zonal statistics. This enabled the visualisation and comparison of behaviour-specific spatial concentrations across the different grazing zones.

Subsequently, spatial hotspot analysis was conducted for each grazing-intensity gradient to examine the clustering patterns of the five behavioural types. The Getis-Ord Gi* statistic was used to identify statistically significant hotspots of behavioural intensity across the study area. Additionally, the global Moran’s I statistic was calculated for each behaviour under different GIGs to evaluate overall spatial clustering patterns. All spatial analysis procedures, including grid generation, zonal statistics, hotspot detection, and spatial autocorrelation, were conducted in ArcGIS Pro 3.3.0 (Esri, Redlands, CA, USA)

### 2.7. Behavioural Proportion Analysis and Peak Fitting of Grazing Behaviour

For the temporal analysis, the focus was on foraging behaviour, so a binary behavioural time series (grazing vs. non-grazing) was constructed within each day. To enable comparative analysis of behavioural probabilities under different grazing regimes and intensities, we divided the data into four segments within our grazing experimental period. This temporal stratification allowed us to visually characterise the distribution patterns of livestock grazing behaviours and assess the influence of management practices and GIGs. By applying Gaussian curve fitting to the temporal distribution of grazing occurrences across the four daily intervals, we identified the central tendency and dispersion of grazing activity. This peak fitting approach allowed for the detection of potential shifts in the timing of grazing peaks across different GIGs.

## 3. Results

### 3.1. Optimal Machine Learning Classifier

After balancing, the five behaviours were more evenly distributed, each accounting for approximately 20%. Compared to the original dataset, the Cluster Centroids undersampling strategy significantly reduced the total sample size, while the SMOTE, ADASYN, and SMOTE-Tomek oversampling strategies substantially increased it. The SMOTE-ENN resampling method led to a moderate increase in the overall sample size (Table 2 and Table 3).

The performance of each classifier was evaluated separately on datasets from both continuous grazing and rotational grazing systems. Evaluation metrics included overall classification accuracy (CA), $F_{1}$-score, and Kappa coefficient. Among the four models, the KNN classifier consistently achieved the highest performance across all metrics and grazing systems (Table 4), indicating its superior suitability for this study.

To further evaluate the classification performance of the KNN model under different resampling strategies, a range of K values (2–31) and distance metrics (Manhattan, Euclidean, Chebyshev) were tested.

Figure 3 shows that across both grazing systems, the combined resampling strategy SMOTE-ENN consistently yielded the highest accuracy, with peak performance at *K* = 2. Classification accuracy (CA) reached 0.968 under continuous grazing and 0.953 under rotational grazing. In comparison, SMOTE-Tomek, SMOTE, and ADASYN achieved moderate performance (CA ≈ 0.62–0.75), while Cluster Centroids performed the worst (CA ≈ 0.10–0.40).

Figure 4 shows that under three distance metrics, SMOTE-ENN again showed superior performance, with CA ranging from 0.902 to 0.930 across both datasets. SMOTE-Tomek, SMOTE, and ADASYN yielded CAs between 0.639 and 0.765, while Cluster Centroids remained suboptimal (CA = 0.162–0.285). Using Manhattan distance with SMOTEENN produced the best results: CA = 0.930 (continuous grazing) and 0.912 (rotational grazing).

### 3.2. Spatial Distribution Patterns of Behaviours

Significant differences were observed in the spatial distribution patterns of fine-scale cold and hotspots for grazing, resting, lying, rumination, and walking behaviours under different grazing-intensity gradients and grazing management systems across the grazing zones, as identified by the KNN model (Figure 5 and Figure 6). Under continuous grazing, the hotspots for rumination, lying, and standing behaviours exhibited increased spatial dispersion with decreasing GIGs, while the spatial extent of grazing behaviour hotspots remained relatively stable and exhibited a more extensive distribution. In contrast, under rotational grazing, the hotspot areas for grazing and standing behaviours decreased and became more scattered with decreasing GIGs. The spatial distribution of hotspots for rumination and lying behaviours showed minimal variation with changes in GIGs. Notably, under both grazing systems, walking behaviour exhibited clearly defined cold and hotspots at moderate grazing (MG).

#### 3.2.1. Spatial Clustering Pattern

The spatial clustering of cattle behaviours exhibited notable variations under different GIGs and management systems, as indicated by Moran’s I and Z-scores (Figure 7). In both grazing systems, behaviours generally showed positive spatial autocorrelation (Moran’s I > 0, *p* < 0.001), indicating significant clustering across all conditions.

#### 3.2.2. Spatial Clustering of Grazing Behaviour Varies with GIGs

In continuous grazing systems, grazing behaviour showed significantly higher values for the reduction of the GIGs. Spatial clustering increased notably with decreasing GIGs (Moran’s I: 0.195 → 0.383 → 0.301), peaking under moderate grazing (Moran’s I = 0.383, Z = 52.38), indicating the strongest spatial clustering. Clustering weakened under heavy (Moran’s I = 0.195, Z = 26.75) and light grazing (Moran’s I = 0.301, Z = 41.30). In contrast, under rotational grazing, spatial clustering of grazing behaviour remained relatively stable across GIGs (Moran’s I: 0.266 → 0.236 → 0.236), with significantly lower clustering at moderate and light intensities (Z = 36.88 and 36.84, respectively) compared to continuous grazing.

#### 3.2.3. Spatial Clustering of Non-Grazing Behaviours Varies with GIGs

Under both continuous and rotational grazing systems, lying behaviour exhibited the highest spatial clustering under heavy grazing (HG). In the continuous grazing system, lying and ruminating behaviours showed the strongest spatial clustering, which declined significantly with decreasing GIGs (Moran’s I for lying: 0.378 → 0.271 → 0.187; ruminating: 0.333 → 0.284 → 0.202). In the rotational grazing system, lying and walking behaviours exhibited the highest spatial clustering, both of which also declined as GIGs decreased (Moran’s I for lying: 0.416 → 0.271 → 0.185; walking: 0.372 → 0.347 → 0.254).

#### 3.2.4. Variance Patterns of Moran’s I for Five Behaviours Across GIGs

As shown in Table 5, under HG, the Moran’s I values for different behaviours varied more in both continuous and rotational grazing systems, with lower variability observed in the rotational system (variance of Moran’s I: 0.00689) compared to the continuous system (0.00766). Under moderate grazing (MG), the variance of Moran’s I between behaviours was similar between systems (continuous: 0.00213; rotational: 0.00293). Under a light grazing intensity, the variance further decreased in both systems (continuous: 0.00202; rotational: 0.00178).

### 3.3. Temporal Probability Distribution Patterns of Behaviours

Grazing behaviour accounted for the highest proportion among all behaviours across both grazing systems. As shown in the Figure 8a, under the rotational grazing system, the proportions of the five behaviours tended to converge with decreasing GIGs. In contrast, under the continuous grazing system (Figure 8b), grazing behaviour remained dominant at moderate and light grazing (MG: 33.56%; LG: 37.55%).

As GIGs decreased, walking behaviour showed a trend of first increasing and then decreasing in both systems (continuous: 8.47–19.59–16.00%; rotational: 7.15–20.69–18.91%). Standing behaviour consistently increased with decreasing GIGs (continuous: 9.33–18.05–20.25%; rotational: 13.31–21.75–24.26%). The proportion of lying behaviour was relatively stable across GIGs, with consistently higher values in the rotational system compared to the continuous system.

### 3.4. Variation in Grazing Behaviour Peaks

The peak fitting analysis of grazing behaviour revealed distinct temporal patterns across GIGs and grazing management systems (Figure 9). Under the continuous grazing system, grazing peaks were more concentrated in the early stages of the heavy grazing stage, with a total of 7 peaks. At moderate grazing stage, the peaks were more evenly distributed over time, totalling 8, while at the light grazing stage, the number of peaks decreased to 4. In the rotational grazing system, the distribution of grazing peaks was more uniform across the GIGs. The number of peaks responded to a gradual decrease as the GIG decreased: 9 (heavy), 7 (moderate), and 3 (light).

## 4. Discussion

Although SMOTE-ENN achieved the highest classification accuracy across both rotational and continuous grazing datasets in this study, its superiority may not be universally applicable. This performance can be attributed to the synergistic effect of SMOTE, which generates synthetic instances for minority classes, and Edited Nearest Neighbours (ENN), which filters out noisy or borderline samples from the majority class. Together, these techniques not only improve class balance but also enhance the definition of class boundaries, leading to better generalisation of the model. However, it is worth noting that the classification accuracy of SMOTE-Tomek was also comparably high, suggesting that alternative resampling strategies may yield competitive or even superior results under different circumstances. In behaviour monitoring tasks with substantially different data characteristics, such as variations in class distribution and sample size, the effectiveness of a given resampling technique is highly context-dependent [29,30]. Additionally, factors, such as the choice of classifier, feature representation, and evaluation metrics, can further influence the performance of resampling approaches [31]. Thus, the effectiveness of resampling methods should be re-assessed in similar grazing behaviour studies.

Since the grazing experiment was designed to simulate a confined, fenced livestock system typical of tropical paddocks, the spatial extent of the study area was relatively limited, and the DSM revealed minimal topographic variation. As a result, the natural environmental conditions across our experimental plots can be considered approximately uniform, which allowed us to effectively control terrain-related variability and focus more directly on the relationship between grazing management systems and the spatiotemporal responses of livestock behaviours. Additionally, since GPS coordinates themselves contain spatial positioning information—including elevation, slope, and aspect—explicitly adding these topographic variables would have introduced multicollinearity [32], potentially reducing the robustness and interpretability of our machine learning models. Moreover, the primary focus of this study was to assess the effects of grazing intensity gradients (GIGs) and grazing management regimes on spatial aggregation patterns and the temporal distribution of cattle behaviour. Following the principle of controlling extraneous variables, we intentionally idealised the terrain-related factors and excluded them from the analysis to avoid unnecessary confounding effects. However, in free-range or heterogeneous mountainous systems, diverse environmental covariates are necessary to capture spatial complexity. Additionally, behaviour misclassification remains a source of error. False positives in behaviour classification can be interpreted as analogous to Type I errors. For example, mistaking ‘rumination’ for ‘standing’ inflates rumination estimates, impacting precision, recall, and possibly creating false spatial hotspots due to prediction bias. Thus, the distinction between inactive behaviours, such as standing, lying, and rumination, remains a limitation in this study. However, recent advancements, such as integrated video monitoring systems [16] and the use of activity sensors, in combination with GPS tracking offer promising solutions to improve the accuracy of behaviours classification in future research.

Spatial hotspot analysis revealed pronounced spatial heterogeneity in behaviours under both continuous and rotational grazing systems, with varying responses to GIGs. In the continuous grazing system, as GIGs decreased, resting behaviours, such as ruminating, lying, and standing, exhibited increasingly dispersed hotspots, indicating a reduction in spatial clustering. Grazing behaviour maintained relatively stable and concentrated spatial hotspots regardless of the change in GIGs. This suggests that, under continuous grazing, high grazing pressure coupled with limited vegetation resources compels cattle to maintain a highly stable spatial distribution of grazing behaviour, concentrated in areas with abundant forage. In contrast, resting behaviours, such as rumination, lying, and standing, tend to exhibit greater spatial dispersion and variability. As GIGs decrease, these low-energy behaviours become more broadly and evenly distributed, whereas grazing activity remains spatially clustered to sustain energy intake and meet physiological demands. This behavioural differentiation aligns closely with principles of energy allocation in livestock, whereby animals prioritise energy-efficient strategies under varying resource availability [33,34]. Under rotational grazing, reduced grazing pressure and increased resource availability lead to a more effective spatial dispersion and balance of grazing hotspots with decreasing GIGs. This suggests that rotational management promotes more flexible spatial behaviour and potentially mitigates overutilisation of localised forage patches.

Moran’s I further quantified these spatial dynamics. Notably, spatial clustering did not respond linearly to changes in GIGs. Under continuous grazing, as GIGs decreased, grazing behaviour under MG and LG exhibited relatively higher Moran’s I values, indicating stronger spatial clustering, whereas high grazing intensity (HG) showed a lower Moran’s I. This pattern may be attributed to intensified intraspecific competition [35] under high stocking pressure, where limited availability of high-quality forage compels cattle to expand their foraging range in search of resources. Such behavioural adjustments likely reflect underlying adaptive mechanisms, in line with previous findings on cattle foraging strategies under resource scarcity [36]. Although the spatial clustering of grazing behaviour appears reduced under HG, the resulting dispersed yet frequent and intense grazing activity—driven by forage scarcity and interspecific competition—can still exert substantial cumulative pressure on the grassland. Consequently, such conditions may accelerate widespread grassland degradation despite the lower observed spatial clustering. Under rotational grazing, spatial clustering for grazing remained low and stable across intensities (Moran’s I = 0.236–0.266), with lower Z-scores compared to continuous grazing. This suggests more spatially dispersed and balanced cattle activity. Moreover, rotational management moderated the clustering of lying, walking, and standing behaviours, which exhibited a consistent decline in spatial aggregation with reduced grazing pressure. These results suggest that rotational grazing increases behavioural sensitivity to changes in GIGs and more effectively mitigates the spatial concentration of high-impact behaviours such as grazing, allowing better control of grassland degradation through stocking rate adjustments.

Analysis of behavioural variance via Moran’s I showed that spatial heterogeneity among behaviours was highest under heavy grazing, particularly in continuous systems. As GIGs decreased, behavioural differences in spatial clustering diminished in both systems, with rotational grazing exhibiting more uniform and consistent behavioural spatial patterns across all intensity levels. This further illustrates the highly responsive nature of behaviour under rotational grazing systems.

On a temporal scale, grazing behaviour remained predominant across both systems. However, rotational grazing promoted a more balanced temporal distribution of behaviours with decreasing grazing intensity, suggesting enhanced behavioural plasticity in response to improved resource availability—a response pattern supported by previous studies [37,38]. In contrast, under continuous grazing, grazing consistently dominated even at medium and light intensities (MG: 33.56%; LG: 37.55%), indicating the reduced responsiveness of behavioural allocation to pressure gradients. In both grazing systems, standing behaviour increased with reduced stocking pressure, likely reflecting decreased foraging urgency, while walking behaviour exhibited a unimodal response, potentially driven by trade-offs between search effort and energy conservation -a behavioural mechanism consistent with previous theoretical frameworks [39]. Notably, peak fitting of grazing behaviour revealed more structured temporal rhythms under rotational management: although the number of grazing peaks decreased from 9 to 3 as intensity declined, their distribution became more uniform, reflecting a stable activity rhythm shaped by rotational strategies. In contrast, continuous grazing exhibited a less coherent pattern, with the peak number and intensity fluctuating more erratically. These findings highlight the role of rotational grazing in modulating both the intensity and temporal structure of high-impact behaviours—a behavioural mechanism discussed by Cheleuitte-Nieves et al. [40].

In the heterogeneous mountainous rangelands of China—such as Qinghai, Inner Mongolia, and Xinjiang—free-range grazing remains a predominant livestock management strategy, allowing animals to roam vast natural pastures at minimal direct cost. These regions are characterised by vast pastoral areas, complex terrain structures, and strong environmental gradients (e.g., elevation, slope, aspect), all of which substantially influence livestock movement patterns and behavioural adaptations [41,42,43]. This practice is deeply intertwined with complex ecosystems and is often implemented across landscapes characterised by steep slopes, varying aspects, and substantial elevation gradients. Such significant horizontal and vertical movement demands substantial energy expenditure and profoundly shapes cattle behaviour patterns. The pronounced topographic variability in these regions necessitates the integration of multiple environmental and terrain variables when modelling grazing behaviours and spatial distribution in these settings. This study focuses on cattle behavioural patterns under tropical/subtropical lowland plains, fenced grazing systems in southern China. Specifically, we examined how different grazing management regimes interact with gradually decreasing grazing intensity gradients under a complex system state. By leveraging a finer spatial resolution and a daily temporal scale, this research provides a more detailed characterisation of cattle behaviour. The integration of multi-class behaviour classification, spatial clustering metrics, and temporal peak fitting offers an innovative and comprehensive perspective on grazing dynamics. It contributes a foundational baseline for understanding the spatiotemporal responses of livestock under sensor-based monitoring and machine-learning-driven analysis, thereby informing future practical management strategies. Building upon this baseline, future research could extend the temporal scope of observations and compare grazing experiments across diverse ecological systems. This may involve incorporating a broader set of environmental variables—such as terrain heterogeneity, vegetation indicators and soil moisture—as well as climatic factors including temperature, precipitation, and solar radiation. Such enhancements would allow for a more in-depth exploration of the Spatial Matching Ratio (Equation (8)) between livestock behaviour hotspots and vegetation degradation indicators (e.g., NDVI decline), across varying grazing management systems and intensity gradients.(8)$$Matching\;Ratio=\frac{Area\;(Behaviour\;hotspot\;\cap \;NDVI\;decline\;hotspot)}{Area\;(Behaviour\;hotspot\;\cup \;NDVI\;decline\;hotspot)}$$

## 5. Conclusions

This study compared the behaviours classification accuracy of four machine learning models across different grazing datasets, identifying KNN as the optimal classifier. Through the implementation of five resampling strategies and KNN parameter sensitivity analysis, we proposed a robust method for grazing data mining. Based on the KNN classification results across different grazing-intensity gradients (GIGs), followed by zonal clustering (Moran’s I) analysis and behavioural peak fitting analysis, this study systematically investigated the spatiotemporal distribution of cattle behaviour under varying GIGs and management systems. The results demonstrated that in continuous grazing systems, even when grazing intensity was reduced, the progressive decline in forage availability continued to intensify the spatial clustering of grazing behaviour, thereby increasing the risk of land degradation. Conversely, rotational grazing systems promoted more spatially dispersed and temporally stable grazing behaviour patterns. Additionally, the spatial clustering of other behaviours showed greater responsiveness to changes in GIGs, while the temporal concentration of grazing behaviour declined in accordance with decreasing GIGs. Taken together, these findings suggest that, when combined with appropriate adjustments to GIGs, rotational grazing can effectively balance the spatial and temporal distribution of behaviours, thereby maintaining the stability of the pasture–livestock system.

## Figures and Tables

**Figure 1 sensors-25-04561-f001:**
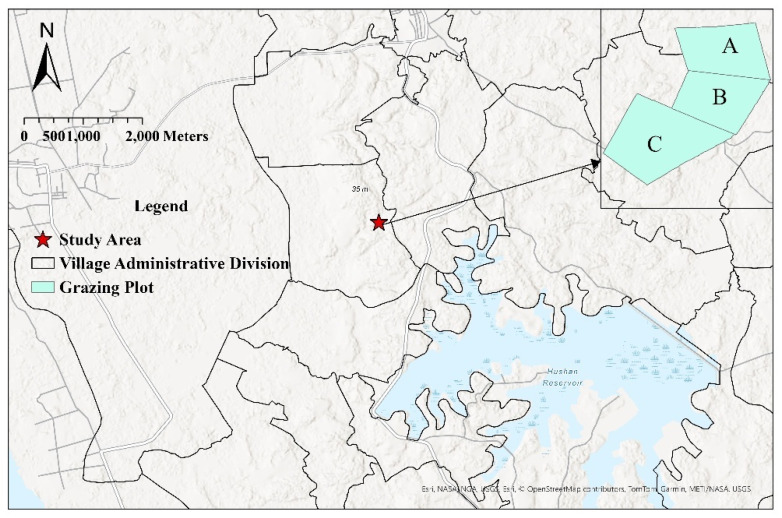
Study area: A, B: rotational grazing plot; C: continuous grazing plot.

**Figure 2 sensors-25-04561-f002:**
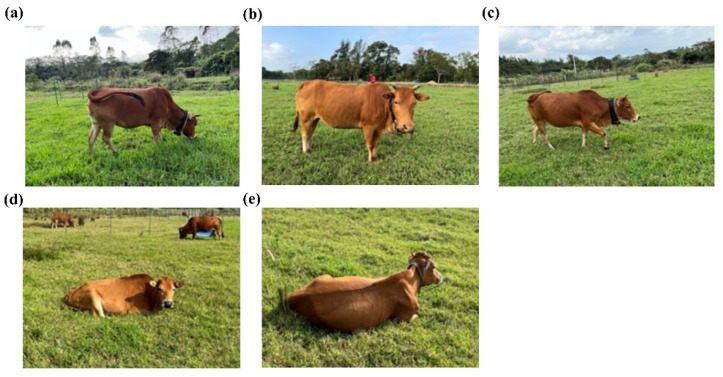
The five main behaviours of cattle: (**a**) grazing, (**b**) standing, (**c**) walking, (**d**) lying, (**e**) rumination.

**Figure 3 sensors-25-04561-f003:**
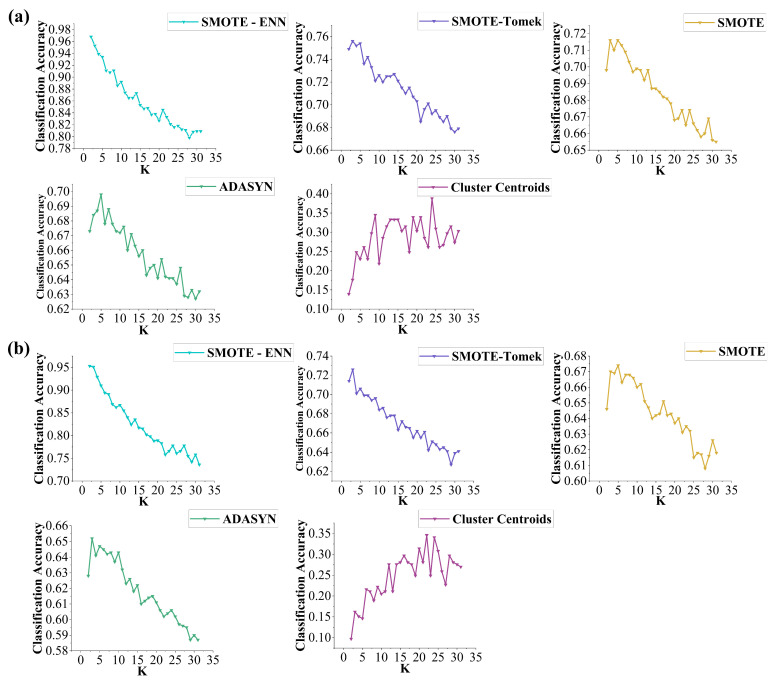
Evaluation of resampling strategies with K value: (**a**) continuous grazing dataset, (**b**) rotational grazing dataset.

**Figure 4 sensors-25-04561-f004:**
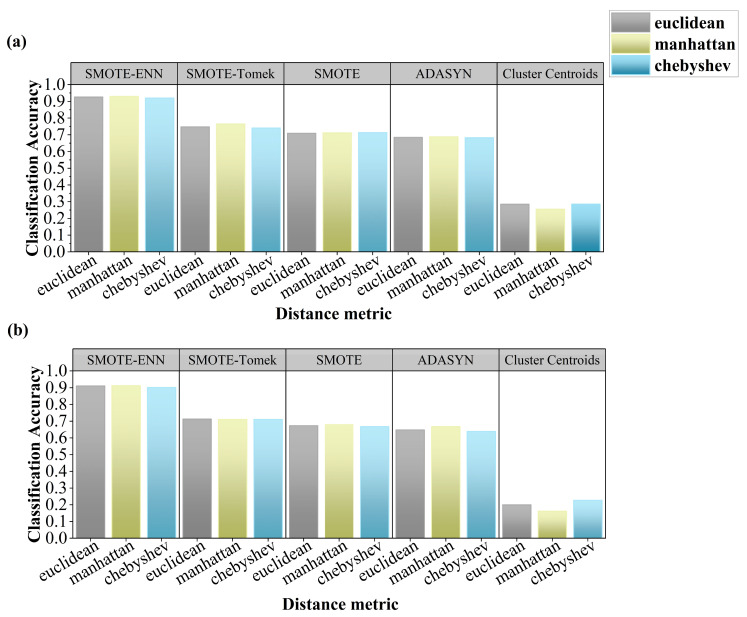
Evaluation of resampling strategies with distance metrics: (**a**) continuous grazing dataset, (**b**) rotational grazing dataset.

**Figure 5 sensors-25-04561-f005:**
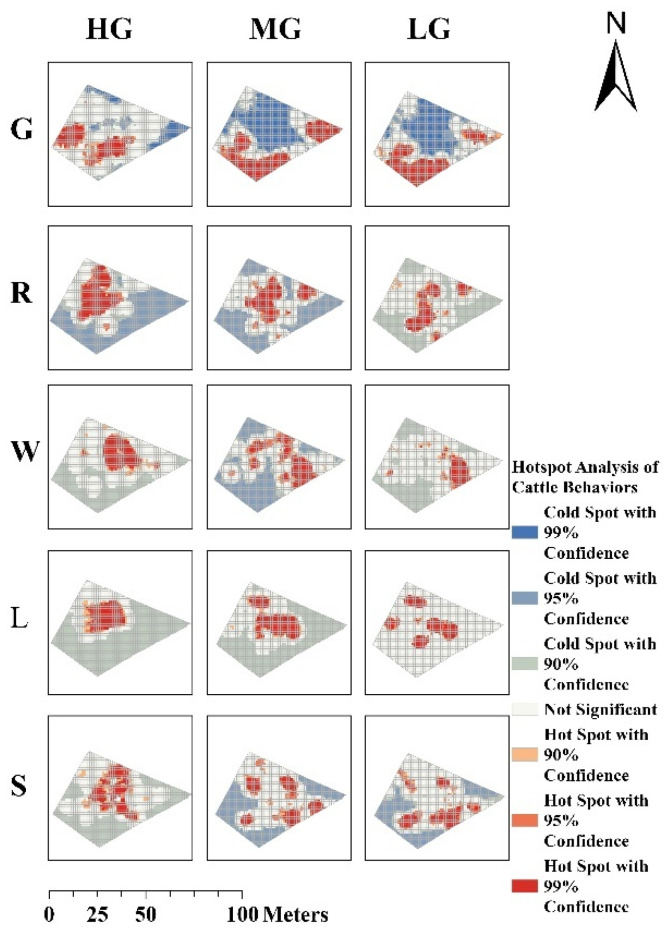
Hot-cold spot maps of five behaviours under continuous grazing (G: Grazing, R: Ruminating, W: Walking, L: Lying, S: Standing; HG: Heavy grazing, MG: Moderate grazing, LG: Light grazing).

**Figure 6 sensors-25-04561-f006:**
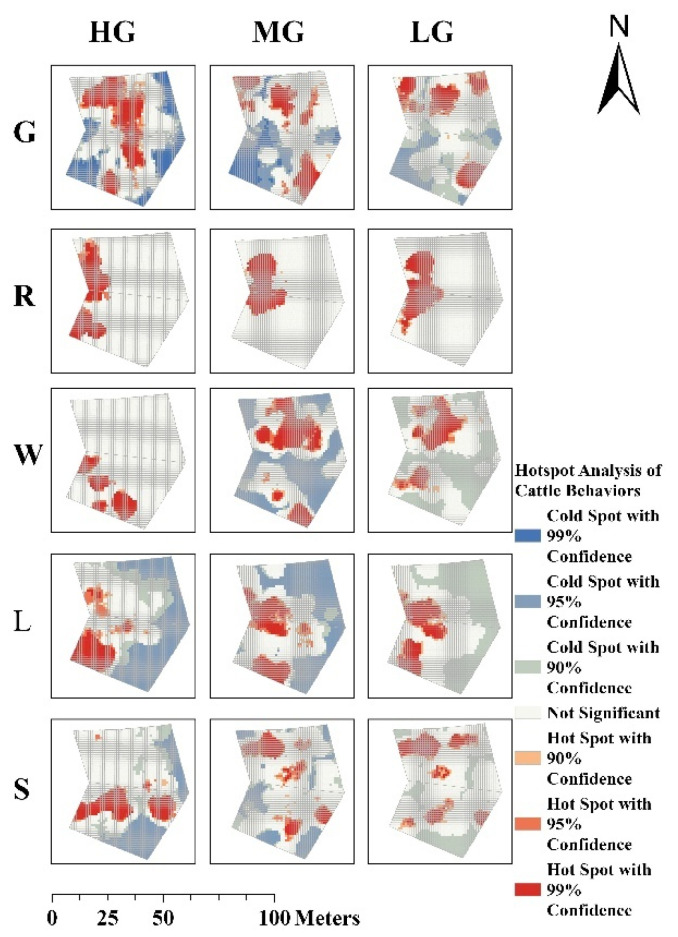
Hot-cold spot maps of five behaviours under rotational grazing (G: Grazing, R: Ruminating, W: Walking, L: Lying, S: Standing; HG: Heavy grazing, MG: Moderate grazing, LG: Light grazing).

**Figure 7 sensors-25-04561-f007:**
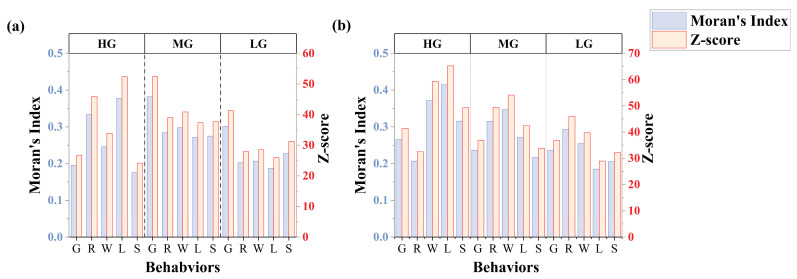
Spatial clustering patterns of cattle behaviours under HG, MG, and LG: (**a**) continuous grazing system, (**b**) rotational grazing system.

**Figure 8 sensors-25-04561-f008:**
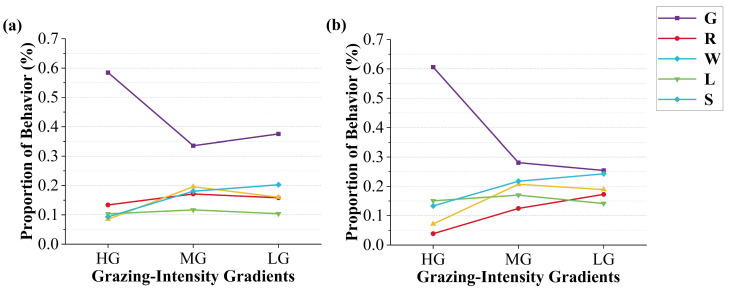
Proportion of time spent on five livestock behaviours vary across HG, MG, and LG: (**a**) continuous grazing system, (**b**) rotational grazing system.

**Figure 9 sensors-25-04561-f009:**
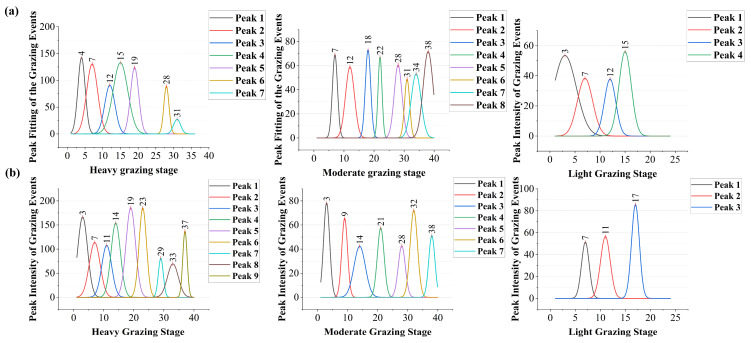
Temporal concentration of grazing behaviour varies across GIGs: (**a**) continuous grazing system, (**b**) rotational grazing system.

**Table 1 sensors-25-04561-t001:** Confusion matrix showing the classification performance across predicted and actual labels.

Actual/Predicted	Positive (P′)	Negative (N′)	Total
Positive (P)	True Positive (TP)	False Negative (FN)	P
Negative (N)	False Positive (FN)	True Negative (TN)	N
Total	P′	N′	P + N

**Table 2 sensors-25-04561-t002:** Data balancing of cattle behaviours under continuous grazing.

Sample Size Distribution Before and After Resampling						
Resampling Strategy	Grazing	Rumination	Lying	Standing	Walking	Total
Unbalanced	4361	566	394	195	110	5626
Cluster Centroids	110	110	110	110	110	550
SMOTE	4361	4361	4361	4361	4361	21,805
ADASYN	4361	4422	4304	4415	4347	21,849
SMOTE-ENN	1970	2061	2132	2217	2375	10,755
SMOTE-Tomek	4361	566	394	195	110	5626

**Table 3 sensors-25-04561-t003:** Data balancing of cattle behaviours under rotational grazing.

Sample Size Distribution Before and After Resampling						
Resampling Strategy	Grazing	Rumination	Lying	Standing	Walking	Total
Unbalanced	5271	458	1022	658	123	7532
Cluster Centroids	123	123	123	123	5269	615
SMOTE	5269	5269	5269	5269	5269	26,345
ADASYN	5269	5356	5398	5269	5253	26,572
SMOTE-ENN	1579	2923	2142	2215	2741	11,600
SMOTE-Tomek	4580	4759	4669	4613	4758	23,379

**Table 4 sensors-25-04561-t004:** Classification performance metrics across different models under continuous and rotational grazing systems.

	Naive Bayes		Random Forest		KNN		XGBoost	
	Continuous	Rotational	Continuous	Rotational	Continuous	Rotational	Continuous	Rotational
CA	0.402	0.402	0.769	0.690	0.965	0.953	0.917	0.923
F_1_-score	0.367	0.344	0.770	0.678	0.965	0.953	0.917	0.923
Kappa	0.251	0.223	0.711	0.604	0.956	0.940	0.896	0.903

**Table 5 sensors-25-04561-t005:** Variance of Moran’s I values among cattle behaviours.

	HG	MG	LG
continuous	0.00766	0.00213	0.00202
rotational	0.00689	0.00293	0.00178

## Data Availability

The data presented in this study are available on request from the corresponding author through appropriate channels.

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
