# Peer review of "Spatiotemporal Mapping of Grazing Livestock Behaviours Using Machine Learning Algorithms"

_sensors, 2025, doi:10.3390/s25154561_

Round 1
Reviewer 1 Report
Comments and Suggestions for Authors
This manuscript investigated the spatiotemporal patterns of livestock behaviours under different grazing management systems and grazing-intensity gradients (GIGs) in Wenchang, China, using high-resolution GPS tracking data and machine learning classification. the K-Nearest Neighbours (KNN) model combined with SMOTE-ENN resampling achieved the highest accuracy. Then the conclusions were reached. The manuscript was well-structured and the research methods and procedures were presented in a relatively clear manner.
There are some issues to discuss:
1、In section 2.4, the format of formulas and formula numbers is very disorganized.
2、In Section 4 discussion, there are still numerous citations in the manuscript. Does this indicate that the current research results have already been studied by others and conclusions have been reached? The results of this assignment merely verified the previous findings, and the manuscript lacks innovation.
3、The data collected in the manuscript cover three time periods. Data were collected for different grazing methods, and then the behavior of the cattle was analyzed. In fact, the activity patterns of cattle in the pasture are quite diverse. Can the data collected within such a short period represent the typical behavior analysis of cattle under different grazing management and different grazing intensities?
4、The several adopted strategy algorithms in the manuscript all use the original algorithm. Has there been any improvement made based on the accuracy of the data obtained from the actual cattle grazing behavior? How can its reliability be verified?
Reviewer 2 Report
Comments and Suggestions for Authors
Suggested revisions for manuscript ID: sensors-3732750
The manuscript explores the spatio-temporal distribution patterns of livestock behavior under different grazing management systems (continuous grazing and rotational grazing) and grazing intensity gradients (heavy, medium, light). The research results can provide data support for the sustainable management of grasslands. The manuscript has certain reference value, but the following need to be supplemented or explained before publication:
- The manuscript only uses GPS trajectory data and does not analyze the behavioral driving mechanisms in combination with environmental factors (such as vegetation NDVI and soil moisture).
- The experiment lasted only 1.5 months (November-December 2022), which did not cover the impact of seasonal changes on behavior. In addition, each intensity gradient was only 10 days, failing to cover seasonal variations (such as the vegetation growth cycle).
- The method described in the manuscript relies on manual pre-observation to calibrate behaviors, and does not use sensors such as accelerometers to cross-validate the ML classification results, which may introduce subjective errors.
- The experiment is only mentioned to be carried out on flat terrain, without considering the impact of complex terrains such as mountains and hills on behavior. This experimental condition may underestimate the impact of heterogeneity.
- The impact of GPS positioning errors (10 meters) on the accuracy of behavior classification, especially the distinction between stationary behaviors such as "standing" and "lying down", is not fully discussed.
- The lack of comparison with research results in other regions, such as grassland ecosystems, requires verification of the universality of the conclusions.
- The basis for the 1m×1m grid is not explained, which may miss micro-scale behavioral characteristics such as foraging patches.
- The analysis of behavioral peaks is based on 4 time periods per day, which may obscure short-term behavioral dynamics (such as morning and evening foraging peaks).
- It is pointed out that rotational grazing reduces behavioral aggregation, but the direct correlation with grassland degradation, such as vegetation restoration data, is not quantified.
- It is recommended to use a three-line table format for Table 1.
Reviewer 3 Report
Comments and Suggestions for Authors
In the reviewed article, the authors analyzed cattle behavior in meadow ecosystems using GPS and ML algorithms. The article is therefore from the area of ecology, agrotechnology and environmental sciences. Nevertheless, the methods used for data collection (GPS), analysis and classification, and spatial analysis mean that the article can be considered to fit into the subject of the journal.
The authors identify new research gaps, which include: the impact of different sampling strategies on ML classifiers, spatial clusters of behaviors and the temporal distribution of foraging. Although similar ML methods have already been used, it can be considered that the combination of SMOTE-ENN with KNN for spatiotemporal analysis is innovative. I also consider the fact that they are based on real experiments to be a great advantage of the study.
The aim of the work is clearly formulated. It is to select the best classifier of cattle behavior and spatial-temporal analysis of these behaviors under different grazing systems and intensities. The data analysis is correct and provides reliable evidence for the validity of the model. The literature review is relevant and well-reasoned. The structure of the article is logical, paragraphs are legible, and figures and tables are correct.
I have a few comments that I would like the authors to address in the text or provide justification for:
1. The authors fitted “about half of the population” of cattle with GPS collars during each grazing phase, with each phase lasting only 10 days. Such short-term and partial recording may not capture the natural variability of animal behaviour in the longer term or under other weather or seasonal conditions. Isn’t this too limited in size and representativeness of the sample?
2. The GPS devices were accurate to ±10 m, and the authors divided the area into a 1 m×1 m grid for hotspot analysis. Couldn’t this mean that measurement error could lead to misidentification of local clusters of behaviour and artificial ‘blurring’ or ‘sharpening’ of hotspots?
3. Isn’t it a problem that generating artificial samples may induce unnatural spatial or temporal patterns?
4. The authors write that “the best classifier was applied to an independent test set,” but they do not state where this set comes from or whether it was collected in a different place or time. Without this, it is difficult to assess how well the model generalizes beyond the experimental conditions.
5. The experiment was conducted on a uniformly flat, fenced pasture. However, the variability of factors such as slope, pasture quality, or meteorological parameters, which strongly affect cattle behavior in real grazing systems, was not taken into account. In my opinion, the lack of analysis of the impact of environmental conditions is a serious limitation of the study.
It is necessary to address the above issues in the article before further processing and clearly indicate the limitations and assumptions of the research.
Round 2
Reviewer 2 Report
Comments and Suggestions for Authors
There is nothing more to comment on. Thank you for the author's careful revisions.
Reviewer 3 Report
Comments and Suggestions for Authors
The authors responded to all my comments in great detail, providing extensive commentary and making any necessary corrections and additions to the text. I fully accept the changes and recommend the text for publication.